# *NOD*2 Mutation-Associated Case with Blau Syndrome Triggered by BCG Vaccination

**DOI:** 10.3390/children8020117

**Published:** 2021-02-06

**Authors:** Akiko Arakawa, Naotomo Kambe, Ryuta Nishikomori, Akiyo Tanabe, Masamichi Ueda, Chikako Nishigori, Yoshiki Miyachi, Nobuo Kanazawa

**Affiliations:** 1Department of Dermatology, Kyoto University Graduate School of Medicine, Kyoto 606-8507, Japan; arakawa@kuhp.kyoto-u.ac.jp (A.A.); nkambe@kuhp.kyoto-u.ac.jp (N.K.); ymiyachi@kuhp.kyoto-u.ac.jp (Y.M.); 2Department of Dermatology and Allergology, University Hospital, Ludwig-Maximilian-University, D-80337 Munich, Germany; 3Department of Pediatrics, Kurume University, Kurume 830-0011, Japan; rnishiko@med.kurume-u.ac.jp; 4Ophthalmology and Visual Science, Kyoto University Graduate School of Medicine, Kyoto 606-8507, Japan; atanabe@kuhp.kyoto-u.ac.jp; 5Institute for Virus Research, Kyoto University Graduate School of Medicine, Kyoto 606-8397, Japan; mueda@virus.kyoto-u.ac.jp; 6Department of Dermatology, Kobe University Graduate School of Medicine, Kobe 650-0017, Japan; chikako@med.kobe-u.ac.jp; 7Department of Dermatology, Hyogo College of Medicine, Mukogawa-cho 1-1, Nishinomiya, Hyogo 663-8501, Japan

**Keywords:** *NOD*2, Blau syndrome, early-onset sarcoidosis, lichen scrofulosorum, BCG vaccination

## Abstract

We describe a patient who developed multiple granulomatous skin lesions after Bacille de Calmette et Guérin (BCG) vaccination without significant effect by topical corticosteroid, followed by painless cystic tumors on the bilateral knees and hands and inflammatory changes on ophthalmologic examination. A functional mutation in *NOD*2 was detected by a genetic analysis, and he was diagnosed as sporadic Blau syndrome. Since *NOD*2 acts as a sensor for the BCG component, it is possible that BCG vaccination may trigger granuloma formation in Blau syndrome patients with such genetic background.

Early-onset sarcoidosis (EOS), which usually occurs in children under four years old, is quite rare and has a distinct triad of skin, joint, and eye disorders, which are chronically progressive and, in many cases, result in blindness and joint destruction [1]. Skin involvement is usually observed several months prior to extracutaneous manifestations. We examined 10 Japanese cases and revealed that EOS is closely related to *NOD*2 mutations, resulting in constitutive nuclear factor (NF)-κB activation, and EOS is now considered to have the same etiology as Blau syndrome (BS), an autosomal dominant-transmitted systemic granulomatosis [2]. Here we describe a sporadic BS patient with a functional *NOD*2 mutation on whom skin papules developed soon after Bacille de Calmette et Guérin (BCG) vaccination, spreading from the vaccinated site.

## 1. Case Report

The patient was the second child of healthy parents, born by normal delivery following full-term uncomplicated gestation. He received BCG vaccination in his left arm at the age of six months, and one month later, non-pruritic solid papules appeared at the vaccinated site. Yellowish-orange asymptomatic papules, 1–3 mm in diameter with some scales (Figure 1A), then spread to involve almost the whole body. Treatment with topical corticosteroids had no significant effect. Although lichen scrofulosorum (LS) was suspected, the tuberculin reaction remained negative. At the age of 18 months, painless cystic tumors appeared on his ankles, and then similar tumors gradually developed on the bilateral knees and hands (Figure 1B), without bone change on X-ray examination. Joint fluid obtained from the ankle was negative for mycobacterium by culture and polymerase chain reaction. Multiple nodules on the pupil margins where the iris adheres to the lens and inflammatory changes on the peripheral retina were observed by ophthalmologic examination.

By histological examination of the skin papules, non-caseating epithelioid cell granulomas accompanied with some multinucleated giant cells were observed mainly in the upper dermis around hair follicles (Figure 2A,D). In order to exclude the possibility of Langerhans cell histiocytosis, immunohistochemical studies were performed to reveal that the epithelioid cells were negative for LAG antibody, specific for Langerhans histiocytes (Figure 2B, Langerhans cells in the follicular epidermis were positive), and positive for BerMac antibody, specific for macrophages (Figure 2C).

At the age of three years, laboratory tests demonstrated a slight elevation of angiotensin-converting enzyme up to 13.0 nmol (normal range for age: 3.4 to 11.0), IgG to 1575 mg/dL (normal: 900 to 1160), IgA to 278 mg/dL (normal: 66 to 120), IgM to 169 mg/dL (normal: 60 to 120), IgE to 566 mg/dL (normal: below 400), and a erythrocyte sedimentation rate to 26 mm per hour (normal: below 12). The peripheral lymphocyte population analyzed by flow cytometry and lymphocyte stimulation with phytohemagglutinin was normal. The following tests were also within the normal range: blood count, urinalysis, liver proteins, serum calcium, antinuclear antibody, chest and skull X-ray, electrocardiogram, and abdominal and cardiac sonograms. One cm subcutaneous lymph nodes with a smooth surface were palpable in the bilateral groins, but no organomegaly was detectable.

After obtaining informed consent, which was approved by the ethics committees at Kyoto University (G-68), a genomic DNA sample was extracted from peripheral blood to determine the nucleotide sequence of all 12 exons of the *NOD*2 gene. As described elsewhere, a 1813A>C (T605P in amino acid change) mutation was discovered and significant promotion of the basal NF-κB activity by this mutation compared with the wild-type *NOD*2 was shown by the luciferase assay [2].

Based on these clinical, histological, and genetic manifestations, the patient was diagnosed as BS. Oral corticosteroids were effective in controlling the progression of uveitis but were difficult to taper off. Development of another *NOD*2-associated disorder such as Crohn’s disease was not observed.

## 2. Discussion

We described here a Japanese sporadic case of BS that followed BCG vaccination. Since Japan is located in a high-risk area for tuberculosis, all infants negative for the tuberculin test receive a BCG vaccination, which contributes to reduce the number of meningitis cases from tuberculosis in childhood [3]. BCG is also used for adjuvant therapy to promote host resistance against infectious diseases and cancers. N-acetyl muramyl-L-alanyl-D-isoglutamine (muramyl dipeptide, MDP), the major component of peptidoglycan contained in bacterial cell wall, is the minimum adjuvant-active structure in BCG [4].

*NOD*2 is predominantly expressed in monocytes and can recognize MDP to activate host defense through NF-κB [5]. *NOD*2 mutations were reported to be associated with inflammatory granulomatous disorders, such as Crohn’s disease, a common inflammatory bowel disease [6], and BS and EOS, rare autosomal dominant or sporadic disorders characterized by arthritis, skin rash, and uveitis [7,8,9]. We have found five additional BS cases in the Japanese cohort who developed skin papules soon after BCG administration at the injected site [10]. Considering that *NOD*2 recognizes MDP as its ligand, BCG vaccination could be one of the common exogenous triggers of granuloma formation in BS patients with functional *NOD*2 mutations. However, it is always difficult to prove the causes of the disease, and further studies are needed to clarify the role of BCG vaccination on the development of BS.

LS, a rare clinical entity classified as tuberculid, which is considered not to be a true mycobacterium infection but an allergic reaction to mycobacterium, was first suspected in our patient. Cutaneous features observed in lichen scrofulosorum are similar to those in BS: maculopapular lesion, superficial dermal granulomas surrounding hair follicles, usually without apparent caseation. Interestingly, LS-like eruption has been reported to develop after BCG vaccination [11]. Considering that MDP, the *NOD*2 ligand, is the common component in BCG, a *NOD*2 activation may take some part in the pathogenesis of LS, similar to the case of BS. Single nucleotide polymorphisms, which are known to be present in *NOD*2 with various NF-κB activations, may genetically decide who will be affected with rare LS among tuberculosis patients and BCG-vaccinated children.

Although BS patients can be controlled with low-dose corticosteroids, severe complications such as blindness, growth retardation, heart involvement, renal failure, joint contractures, and even death can develop, as previously described in EOS cases [10,12,13]. A recent report has suggested the usefulness of early intervention with biologics such as anti-TNF drugs to prevent the irreversible changes [10].

Here we described a sporadic BS patient with a functional *NOD*2 mutation who developed skin papules after BCG vaccination. This case provides a possible model of the pathogenesis of BS in which the local immune response to MDP triggered granuloma formation in a genetically predisposed host.

## Figures and Tables

**Figure 1 children-08-00117-f001:**
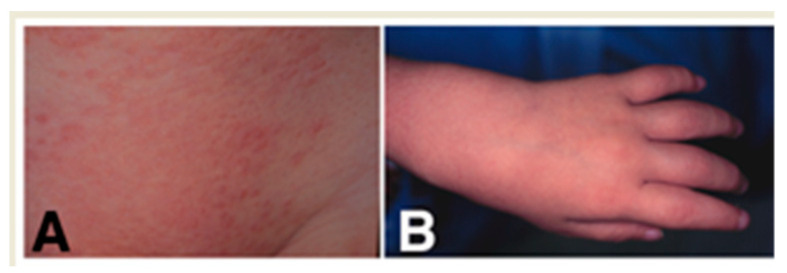
Clinical appearance. Yellowish-orange asymptomatic papules in the trunk are shown (**A**). Phalangeal joints are swelling (**B**).

**Figure 2 children-08-00117-f002:**
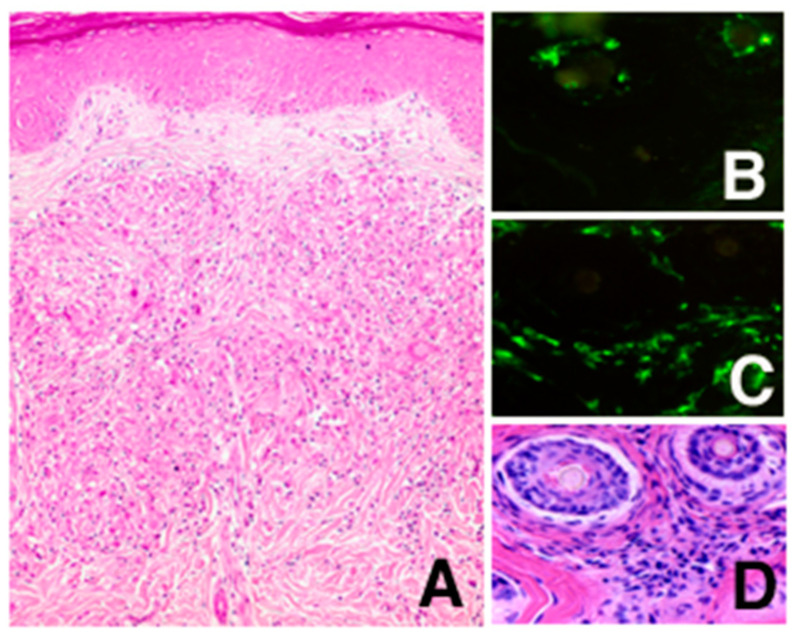
Histological findings. A section of the skin papule was stained with hematoxylin and eosin (**A**). To characterize the infiltrating epithelioid cells, sequential sections were stained with Langerhans cell-specific LAG antibody (**B**), macrophage-specific BerMac antibody (**C**), and hematoxylin and eosin (**D**). (Original magnification. **A**, ×200; **B**–**D**, ×400).

## Data Availability

The data presented in this study are available on request from the corresponding author. The data are not publicly available due to patient’s privacy.

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
