# Peer review of "NOD2 Mutation-Associated Case with Blau Syndrome Triggered by BCG Vaccination"

_children, 2021, doi:10.3390/children8020117_

Round 1

Reviewer 1 Report

In the study of Arakawa and coworkers the authors present a patient with sporadic Blau syndrome triggered by BCG vaccination. This is a very interesting case however the authors are requested to ameliorate their work by clearly reporting the diagnosis of BS before the discussion section in the "case report" section and provide some information about the management and the outcome of the patient. A short discussion about the management and outcome of BS should also be performed in the discussion section.

Author Response

Dear Reviewer 1,

In the study of Arakawa and coworkers the authors present a patient with sporadic Blau syndrome triggered by BCG vaccination. This is a very interesting case however the authors are requested to ameliorate their work by clearly reporting the diagnosis of BS before the discussion section in the "case report" section and provide some information about the management and the outcome of the patient. A short discussion about the management and outcome of BS should also be performed in the discussion section.

> We appreciate your supporttive comments on our report.

According to your suggestion, descriptions on the diagnosis and the clinical course of the patient have been added in Case Report and the management and outcome of BS have concisely been discussed in Discussion.

Reviewer 2 Report

This is an interesting case and worth reporting. It requires some minor edits to improve the manuscript.

Abstract

Essentially the case report describes how this child was identified to have Blau syndrome. The trigger to screen for the rare disorder was the skin lesions that developed after the BCG vaccination, and following little effect from treatment, investigations eventually lead to a genetic diagnosis of NOD2 mutation. This abstract should be rewritten to make this clearer.

Case report

This is written well and leads the reader through the steps that led to the genetic screening. It would be good to include what the clinical team offered this patient after diagnosis such as eventual treatment and screening for other NOD2 associated inflammatory disorders.

Discussion

It is understood both Blau and EOS are generally grouped into a set of juvenile granulomatous systemic disorders and both share the same genetic mutations. However, this should be made clear in the manuscript, since the authors skip between both terms and this may confuse the reader. The authors also mention that the 1813A>C (T605P) mutation is novel. However, this has been reported previously in Blau syndrome (see ref below). Please can the authors check this again.

Overall, it is always difficult to prove cause and effect in these cases. Did the BCG trigger the skin lesions? Having read through some of the literature, Blau syndrome has previously been diagnosed after BCG vaccination in other case series. It would be worth nothing this in the discussion. I have included some useful references that might help.

Useful References

Kanazawa, N., et al. (2005). "Early-onset sarcoidosis and CARD15 mutations with constitutive nuclear factor-κB activation: common genetic etiology with Blau syndrome." Blood 105(3): 1195-1197

Matsuda T, Kambe N, Ueki Y PIDJ members in the JSIAD, et al Clinical characteristics and treatment of 50 cases of Blau syndrome in Japan confirmed by genetic analysis of the NOD2 mutation Annals of the Rheumatic Diseases 2020;79:1492-1499.

Caso F, Galozzi P, Costa L, et al. Autoinflammatory granulomatous diseases: from Blau syndrome and early-onset sarcoidosis to NOD2-mediated disease and Crohn’s disease. RMD Open 2015;1:e000097. doi:10.1136/rmdopen-2015- 000097

Punzi L, Furlan A, Podswiadek M, Gava A, Valente M, De Marchi M, Peserico A. Clinical and genetic aspects of Blau syndrome: a 25-year follow-up of one family and a literature review. Autoimmun Rev. 2009 Jan;8(3):228-32. doi: 10.1016/j.autrev.2008.07.034. Epub 2008 Aug 19. PMID: 18718560.

Author Response

Dear Reviewer 2,

This is an interesting case and worth reporting. It requires some minor edits to improve the manuscript.
Abstract
Essentially the case report describes how this child was identified to have Blau syndrome. The trigger to screen for the rare disorder was the skin lesions that developed after the BCG vaccination, and following little effect from treatment, investigations eventually lead to a genetic diagnosis of NOD2 mutation. This abstract should be rewritten to make this clearer.

> We appreciate your supportive comments.

According to your suggestion, Abstract has been rewritten to clarify how this patient was identified to have BS.

Case report
This is written well and leads the reader through the steps that led to the genetic screening. It would be good to include what the clinical team offered this patient after diagnosis such as eventual treatment and screening for other NOD2 associated inflammatory disorders.

> We appreciate your supportive comments.

According to your suggestion, the diagnosis and the clinical course of the patient was added to Case Report.

Discussion
It is understood both Blau and EOS are generally grouped into a set of juvenile granulomatous systemic disorders and both share the same genetic mutations. However, this should be made clear in the manuscript, since the authors skip between both terms and this may confuse the reader. The authors also mention that the 1813A>C (T605P) mutation is novel. However, this has been reported previously in Blau syndrome (see ref below). Please can the authors check this again.
Overall, it is always difficult to prove cause and effect in these cases. Did the BCG trigger the skin lesions? Having read through some of the literature, Blau syndrome has previously been diagnosed after BCG vaccination in other case series. It would be worth nothing this in the discussion. I have included some useful references that might help.

Useful References
Kanazawa, N., et al. (2005). "Early-onset sarcoidosis and CARD15 mutations with constitutive nuclear factor-κB activation: common genetic etiology with Blau syndrome." Blood 105(3): 1195-1197
Matsuda T, Kambe N, Ueki Y PIDJ members in the JSIAD, et al Clinical characteristics and treatment of 50 cases of Blau syndrome in Japan confirmed by genetic analysis of the NOD2 mutation Annals of the Rheumatic Diseases 2020;79:1492-1499.
Caso F, Galozzi P, Costa L, et al. Autoinflammatory granulomatous diseases: from Blau syndrome and early-onset sarcoidosis to NOD2-mediated disease and Crohn’s disease. RMD Open 2015;1:e000097. doi:10.1136/rmdopen-2015- 000097
Punzi L, Furlan A, Podswiadek M, Gava A, Valente M, De Marchi M, Peserico A. Clinical and genetic aspects of Blau syndrome: a 25-year follow-up of one family and a literature review. Autoimmun Rev. 2009 Jan;8(3):228-32. doi: 10.1016/j.autrev.2008.07.034. Epub 2008 Aug 19. PMID: 18718560.

> We appreciate your supportive comments.

To avoid confusion, descriptions on EOS have been rewritten in Discussion and the term “novel” has been deleted.

I agree to your comment that it is always difficult to prove the cause of the disease in these cases and have added some comments on it in Discussion.

Some descriptions on the management and outcome of BS have been added with recommended references.